# Remodeling of Bone Marrow Niches and Roles of Exosomes in Leukemia

**DOI:** 10.3390/ijms22041881

**Published:** 2021-02-13

**Authors:** Takanori Yamaguchi, Eiji Kawamoto, Arong Gaowa, Eun Jeong Park, Motomu Shimaoka

**Affiliations:** 1Department of Molecular Pathobiology and Cell Adhesion Biology, Mie University Graduate School of Medicine, 2-174 Edobashi, Tsu-City, Mie 514-8507, Japan; t-yamaguchi@clin.medic.mie-u.ac.jp (T.Y.); a-2kawamoto@med.mie-u.ac.jp (E.K.); arong-g@doc.medic.mie-u.ac.jp (A.G.); epark@doc.medic.mie-u.ac.jp (E.J.P.); 2Department of Hematology and Oncology, Mie University Graduate School of Medicine, 2-174 Edobashi, Tsu-City, Mie 514-8507, Japan; 3Department of Emergency and Disaster Medicine, Mie University Graduate School of Medicine, 2-174 Edobashi, Tsu-City, Mie 514-8507, Japan

**Keywords:** leukemia, exosomes, bone marrow niches

## Abstract

Leukemia is a hematological malignancy that originates from hematopoietic stem cells in the bone marrow. Significant progress has made in understanding its pathogensis and in establishing chemotherapy and hematopoietic stem cell transplantation therapy (HSCT). However, while the successive development of new therapies, such as molecular-targeted therapy and immunotherapy, have resulted in remarkable advances, the fact remains that some patients still cannot be saved, and resistance to treatment and relapse are still problems that need to be solved in leukemia patients. The bone marrow (BM) niche is a microenvironment that includes hematopoietic stem cells and their supporting cells. Leukemia cells interact with bone marrow niches and modulate them, not only inducing molecular and functional changes but also switching to niches favored by leukemia cells. The latter are closely associated with leukemia progression, suppression of normal hematopoiesis, and chemotherapy resistance, which is precisely the area of ongoing study. Exosomes play an important role in cell-to-cell communication, not only with cells in close proximity but also with those more distant due to the nature of exosomal circulation via body fluids. In leukemia, exosomes play important roles in leukemogenesis, disease progression, and organ invasion, and their usefulness in the diagnosis and treatment of leukemia has recently been reported. The interaction between leukemia cell-derived exosomes and the BM microenvironment has received particular attention. Their interaction is believed to play a very important role; in addition to their diagnostic value, exosomes could serve as a marker for monitoring treatment efficacy and as an aid in overcoming drug resistance, among the many problems in leukemia patients that have yet to be overcome. In this paper, we will review bone marrow niches in leukemia, findings on leukemia-derived exosomes, and exosome-induced changes in bone marrow niches.

## 1. Introduction

Leukemia is a hematological malignancy originating from bone marrow hematopoietic stem cells. Characteristic chromosomal translocations and genetic mutations have been identified as the etiology of leukemia, and further advances in cytogenetic studies have added new insights into the pathogenesis of leukemia. The revision of the 2016 World Health Organization (WHO) classification of myeloid neoplasms and leukemia is now used as the classification standard for leukemia worldwide [1]. The four main types of leukemia are well known: chronic myelogenous leukemia (CML), chronic lymphocytic leukemia (CLL), acute myelogenous leukemia (AML), and acute lymphocytic leukemia (ALL) (Figure 1).

Acute leukemia accounts for about two-thirds of all leukemia cases, has a low remission rate, and is a major cause of cancer-related death. In addition to conventional chemotherapy, hematopoietic stem cell transplantation (HSCT) has been established, and new therapies such as molecular-targeted therapy and immunotherapy have been developed in succession, resulting in remarkable advances. However, recurrence is still a problem in many cases, and once it occurs, the prognosis is poor with an overall 5-year survival rate of 40% to 50% in AML [2] and an overall 5-year survival rate reaching no more than 10% in ALL [3,4]

Therefore, it is very important to understand not only the causes of leukemia but also the mechanisms of invasion, relapse, and chemotherapy resistance and to develop new treatment strategies and early detection of relapse.

The concept of the bone marrow niche was first proposed by Schofield in 1978 [5]. The bone marrow niche functions through the interaction of various cells such as bone marrow mesenchymal stem and progenitor cells, osteoprogenitor cells, sinusoidal endothelial cells, perivascular stromal cells, adipocytes, unmyelinated Schwann cells, and immune cells, as well as multiple components such as various adhesion factors, growth factors, and chemokines, to regulate cell self-renewal and differentiation [6]. Normal hematopoiesis is tightly regulated by the interaction of bone marrow niches with hematopoietic stem cells (HSCs) through a variety of signaling pathways [7]. The bone marrow niche is divided into the endosteal niche, the vascular niche [8,9], and some authors would add the reticular bone marrow niche as well, dividing it into three niches [10]. On the other hand, Hira et al. proposed a theory that the HSC niche consists of two compartments: a periarteriolar compartment and a perisinusoidal compartment. Indeed, their work focused on two important findings [11]. First, HSCs are in contact with C-X-C motif chemokine ligand 12 (CXCL12)-abundant reticular (CAR) cell processes throughout the bone marrow, which is one of the main components of the reticular niches. In addition, the HSCs coordinate with both the endosteal and perivascular niches and do not function independently of one other. Second, arteries are mainly located near the endosteum and trabecular bone, while sinusoids are mainly located some distance away from both, resulting in differences in intracellular ROS levels and ultimately in the fate of HSCs. Based on these findings, rather than dividing the HSCs into separate niches, they propose to divide them into two compartments: a periarterial compartment that maintains the HSC pool and a perisinusoidal compartment where HSCs differentiate into progenitor cells, proliferate, further differentiate, and migrate out of the HSC niche.

Exosomes are secreted by all cell types, and of course by tumor cells, including leukemia cells [12]. Exosomes can be easily collected from bodily fluids such as blood, urine, saliva, and breast milk, through which they serve as an important cell-to-cell communication tool not only for nearby cells but also for more distant target cells. These effects are carried out by the transport of DNA, RNA, microRNA (miR), proteins, and cytokines contained in exosomes to target cells, and exosomes are known to play an important role as a communication tool between cells [13,14]. Moreover, it is now known that they play multiple roles in immune response, antigen presentation, differentiation, tumor cell invasion, and distant metastasis. Although the roles played by solid tumor-derived exosomes in tumor growth, angiogenesis, metastasis, and chemotherapy resistance have been widely investigated [15,16,17], their role in hematological malignancies, including leukemia, has not yet been fully investigated. Recent studies indicate that exosomes function as important communication tools between leukemic cells, normal hematopoietic cells, and mesenchymal cells and further support the progression of the disease not only by proliferating leukemic cells themselves, but also by changing the surrounding environment to one that is favorable to them [18].

Understanding the remodeling of the bone marrow niche in leukemia and the exosomes involved in this process can lead not only to a better understanding of the pathogenesis of leukemia but also to more accurate and earlier diagnosis, safer and more effective treatments, more accurate evaluations following treatment, and early detection of relapse. Such advances may hold the key to solving the various problems that still plague leukemia patients today. In this review, we will examine the remodeling of bone marrow niches in leukemia and the roles played by exosomes in relation to this mechanism.

## 2. Leukemic Bone Marrow Niche

Leukemia cells interact with the bone marrow niche and are closely associated with leukemia progression, suppression of normal hematopoiesis and chemotherapy resistance, which is precisely the area of ongoing study [19,20,21]. Leukemic cells and hematopoietic stem cells interact in the bone marrow microenvironment to regulate a wide range of cellular functions. This involves not only cell proliferation, differentiation, and migration, but also cell quiescence and clonal expansion [22]. Leukemia cells disrupt the normal bone marrow niche and transform it into a more convenient niche for themselves so that they can grow more favorably than normal hematopoietic stem cells [23], which is a process aptly described as “leukemia cells hijack bone marrow niche” [11]. By hijacking the hematopoietic niche, leukemic cells become leukemic stem cells (LSCs) and proliferate using the same molecular mechanisms as HSCs. This mechanism can similarly be used to therapeutically target molecular pathways that are aberrantly expressed or constitutively activated in LSCs. It is also important to understand this mechanism since the eradication of LSCs is considered essential for the prevention of relapses in leukemia relapse. In fact, therapies that specifically target the major mechanisms of LSCs are expected to provide higher therapeutic efficacy.

### 2.1. The Bone Marrow Niche as an Accomplice to Leukemia

In addition to HSCs, leukemia cells also act on non-hematopoietic components such as bone marrow cells, adipose tissue, mesenchymal and vascular endothelial cells, the immune microenvironment, and the sympathetic nervous system to create an environment in which they can thrive.

Mesenchymal progenitor cells exist near blood vessels in bone marrow, and HSCs are in contact with these cells. Mesenchymal progenitor cells express abundant niche factors, such as CXCL12, which are important for the maintenance of hematopoietic stem cells. Similar to HSC, leukemia stem cells depend on adhesion signals to their supporting niche for their maintenance [24]. The density of the microvasculature has been shown to be higher in the bone marrow of patients with AML, which correlates with the disease’s aggressiveness [25]. This means that understanding the interaction between the vascular niche and leukemic cells is one of the most important aspects to understanding the pathophysiology of leukemia.

In the bone marrow niche, accessory cell types such as osteoblasts, macrophages, and nerve cells often indirectly influence HSCs; in fact, they play important roles in maintaining HSCs, and leukemic cells can affect them. Osteoblasts are one of the important components of the HSC niche, influencing the homing and development of HSCs [26]. Leukemia alters the bone marrow niche by decreasing the number of osteoblasts in the niche, but maintaining these can hinder the progression of leukemia [27,28].

Macrophages do not merely serve as immune cells but also contribute to the regulation of HSCs in the bone marrow niche [29]. In a mouse model of leukemia, it has been shown that leukemia cells maintain LSC functionality by inhibiting macrophages [30]. Moreover, in recent years, the function of tumor-associated macrophages, known as leukemia-associated macrophages, has been attracting attention, and their various roles in leukemia are now becoming better known [31].

For example, it is known that under normal conditions, mesenchymal stromal cells (MSCs) and the sympathetic nervous system combine to regulate HSC functions [32]. However, leukemic cells suppress normal hematopoiesis by disrupting sympathetic nervous system functions. Hanoun et al. have shown that in an AML model mouse, leukemia-induced sympathetic neuropathy causes changes in the hematopoietic HSC niche and promotes leukemia progression through such altered niches [33].

Adipocytes are also one of the important components of the bone marrow niche. They were previously thought to be just negative regulators that inhibited hematopoiesis [34], but it has recently emerged that adipocytes can promote hematopoiesis [35]. Han et al. showed that adipose tissue can act as a reservoir for hematopoietic stem and progenitor cells [36]. Boyd et al. demonstrated that AML cells suppress hematopoiesis to disrupt the adipocytic niche in BM [37]. AML cells can also functionally change the bone marrow adipocytes for one more favorable to themselves. They accomplish this by inducing the transfer of fatty acids from to adipocytes. Then, AML cells use those fatty acids for generating the energy needed for their growth and proliferation [38].

The bone marrow niches are composed of not only stromal cells but also extracellular matrix proteins. Among the latter, fibronectin has been shown to support the survival and proliferation of hematopoietic stem and progenitor cells via very late angiten-4 (VLA-4), which is a member of the β1 integrin family. VLA-4 interacts with the VCAM-1 expressed by stromal cells and serves as an adhesion and signaling receptor [39]. These molecular mechanisms also play an important role in leukemia. For example, in acute myeloid leukemia, they have been shown to support leukemic cell survival via the VLA-4-mediated phosphatidylinositol 3-kinase/Akt/B cell lymphoma 2 (Bcl2) signaling pathway [40]. One example from these related studies involves the expression of VLA-4 during the first relapse in children with ALL. The study by Shalapour et al. showed that high expression of VLA-4 at first relapse is associated with lower relapse-free survival and overall survival, indicating that the high expression of VLA-4 is a marker of poor prognosis [41]. This is because the expression of genes involved in the PI3K/Akt pathway differs in ALL cells with high VLA-4 expression versus those with low VLA-4 expression. These findings suggest that the VLA-4-mediated signaling pathway is involved in leukemic cell survival and in responses to therapy.

Kindlin-3 (K3) is one of the integrin-regulated molecules identified as the causative gene for Kindler syndrome, which is a rare recessive genetic disorder [42]. Integrin adhesion via K3 is an important factor in normal hematopoietic cell homing and bone marrow retention functions in healthy individuals [43] K3 is an essential integrin-binding and integrin-activating adaptor protein [44], whose loss leads to dysfunctional erythrocytes, platelets, and immune effector cells collectively termed leukocyte adhesion deficiency type III in humans [45,46,47,48]. The loss of K3 in a mouse model of CML triggers the release of LSCs from the bone marrow (BM) into the circulation and impairs their retention, proliferation, and survival in secondary organs, which suppresses CML development, progression, and metastatic dissemination. CML-LSCs express cytotoxic T lymphocyte-associated antigen 4 (CTLA-4), but not normal hematopoietic stem cells. Depleting K3 with a CTLA-4–binding RNA aptamer linked to a K3-small interfering RNA in CTLA-4^+^ LSCs induces disease remission and prolongs the survival of mice with CML [49]. Disrupting the interactions of LSCs with the BM environment is a promising strategy to halt the disease-inducing and relapse potential of LSCs.

### 2.2. Leukemic Bone Marrow Niche and Chemotherapy

One of the bottlenecks hampering the treatment of leukemia is cytopenia caused by chemotherapy. The leukemic bone marrow niche is also thought to be associated with this mechanism.

AML cells produce angiogenic factors that remodel the vascular niche in the bone marrow in order to form a niche favorable to leukemia. However, previous studies have not demonstrated that the inhibition of these angiogenic factors leads to improved prognosis in AML patients [50,51]. One reason for this could be that the remodeling of the vasculature in the central bone marrow and endosteal regions takes a different course during the progression of AML. Duarte et al. reported that endosteal vessels were profoundly remodeled, while the central myeloid vessels were preserved. This is related to the fact that Tumor Necrosis Factor (TNF) and C-X-C Motif Chemokine Ligand 2 (CXCL2) levels were locally increased in the endosteal region. AML cells in the endosteal region produce those pro-inflammatory and anti-angiogenic cytokines and degrade bone endothelium, HSCs, stromal cells, and osteoblastic cells. This causes the bone marrow to misfunction and reduces normal hematopoietic function. Approaches that protect the endothelium have the potential to preserve hematopoietic function, prevent serious infections and bleeding due to leukemia progression and chemotherapy-induced cytopenia, and allow for safer chemotherapy [52].

Recently, Alexander et al. reported that MSC-derived stanniocalcin 1 (STC1) and its transcriptional regulator hypoxia inducible factor (HIF)-1α are important factors in the suppression of HSCs. STC1 is found at elevated levels in the peripheral blood and bone marrow in acute myeloid leukemia. Moreover, HIF-1α stabilization in MSCs induces STC1 secretion. In AML, the upregulation of these factors may lead to the suppression of HSCs, which in turn inhibits normal hematopoiesis. The suppression of normal hemopoiesis in patients with leukemia is a crucial factor leading to cytopenia and tumor-related deaths from severe infection and bleeding. Inhibiting HIF-1α and STC1, promoting residual normal hematopoiesis, and reducing cytopenia in AML suggest that this approach may represent a promising option for safely completing chemotherapy [53].

The acquisition of drug resistance is a very important factor in the likelihood of treatment failure, and the involvement of the bone marrow niche in drug resistance mechanisms has also been shown [54].

C-X-C chemokine receptor type 4 (CXCR4) is intimately involved in the homing and maintenance of hematopoietic stem cells in the bone marrow niche. AML cells also express this CXCR4 on their cell surface. SDF-1α (also termed CXCL12) is secreted by bone marrow stromal cells (BMSCs) and activates CXCR4, which is expressed on AML cells and promotes the migration and retention of AML cells in the bone marrow. In fact, CXCR4 may be involved in the mechanisms that allow leukemic cells to escape apoptosis and chemotherapy-induced cell death, leading to residual disease and relapse [55]. Huang et al. focused on this CXCR4/SDF-1α interaction and found that the addition of a CXCR4 peptide antagonist to cytarabine, which is used in conventional chemotherapy in leukemia, has a synergistic effect on AML treatment. This has now emerged as one of the candidates to eradicate residual disease and overcome chemoresistance in AML [56].

C-X-C motif chemokine ligand 12 (CXCL12) is expressed by a wide range of cell types in the bone marrow including osteoblasts, vascular endothelial cells, and mesenchymal stem cells [57]. CXCL12 is the chemoattractant for HSCs and is thought to play an important role in the maintenance of HSCs by not only storing them in the bone marrow but also by maintaining their quiescence and proliferative capacity [58]. Based on this finding, Agarwal et al. reported that CXCL12 deficiency in endothelial cells inhibited LSC proliferation. On the other hand, the deletion of CXCL12 in MSCs markedly reduced LSC quiescence and increased the self-renewal capacity of LSCs [59]. The deletion of CXCL12 in MSC increases LSC elimination by TKI treatment. This suggests that by activating quiescent LSCs and causing them to increase and expand their cell cycle, the effect of TKIs can take hold in those LSCs that were originally inactive to TKIs. CXCL12-expressing MSC niches may be potential targets for enhancing TKI sensitivity. This represents one of the promising therapeutic targets in the treatment of CML, particularly given the problems associated with minimal residual disease (MRD).

Gap junctions are specialized structures that directly link adjacent cytoplasm and are created by the association of connexins (Cxs), six of which come together to form connexon. The cytoplasm of two cells can communicate directly with one other through connexons, and the movement of ions, microRNAs, ATP, and other small metabolites between neighboring cells regulates various cellular functions. Cxs work to regulate intracellular pathways and gene transcription during cell–cell contact [60,61]. The close interaction between leukemia cells and BM mesenchymal stromal cells (BM-MSCs) plays an important role in the acquisition of chemotherapy resistance. Leukemia cells and BM-MSCs are known to interact not only indirectly through exosomes but also directly through Cxs [62]. Kouzi et al. showed that Cx25 gap junctions may be involved in the interaction between AML cells and BM-MSCs in leukemia niches. Carbenoxolone (CBX) inhibits the assembly of gap junctions and has been used to study the Cx blockade. By inhibiting gap junctions, CBX has been reported to synergize the effects of cytarabine to overcome chemoresistance. This indicates that it is possible to develop therapies to overcome the formation of chemoresistance and to eradicate leukemic cells [63].

Three types of (E, L, P)-selectins have been reported to be involved in the adhesion of leukocytes to vascular endothelial cells. Among them, E-Selectin is an important factor in the vascular endothelial niche. The resistance to treatment and relapse of leukemia is thought to be related to a minimal residual of leukemic cells that remain in the bone marrow, where they can escape the killing effects of antileukemic drugs and evade immune surveillance mechanisms, thus creating the conditions for relapse. In AML, leukemic cells attach to the vascular endothelial niche via E-Selectin and hide in the bone marrow, barely surviving as minimal residual disease (MRD) that cannot be detected by clinical tests, which eventually leads to relapse. Runt-related Transcription Factor (RUNX) is involved in the activation of E-selectin and the inhibition of RUNX in leukemia cells, thereby suppressing cell proliferation [64,65,66,67]. Morita et al. have shown that the suppression of E-Selectin by RUNX inhibitors can significantly prolong survival by inhibiting the pooling of leukemic cells in the bone marrow and facilitating the exposure of leukemic cells released into the circulating blood to anticancer therapy, suggesting one possible way to overcome MRD in leukemia [68].

Although dramatic progress has been made in recent years in the diagnosis and treatment of leukemia, conventional chemotherapy that focuses only on leukemia cells had limited efficacy. This is because the interaction between leukemic cells and the bone marrow niche is very important to the pathophysiology of leukemia. Novel therapeutic strategies that not only target leukemia cells but also treat bone marrow niches are expected to improve the prognosis of leukemia patients.

## 3. Leukemic Exosomes and Bone Marrow Niche Remodeling

### 3.1. Leukemic Exosomes

Exosomes have been shown to play an important role in hematological malignancies. Through exosomes, leukemic cells create a bone marrow microenvironment favorable to their own survival by evading tumor immunity, influencing angiogenesis, and suppressing normal hematopoiesis. Exosomes are thought to be an important mediator of communication between leukemic cells and the bone marrow microenvironment [69].

Leukemia cells have a mechanism of exosome-mediated angiogenesis that allows them to proliferate more actively and to lay the groundwork for spreading to distant organs. The bone marrow of patients with CML is known to be hypoxic, although there is marked angiogenesis in the bone marrow [70,71]. It has been reported that exosomes released by chronic myeloid leukemia cells are associated with a mechanism that promotes angiogenesis in human vascular endothelial cells (HUVECs) [72]. Interleukin-8 (IL-8), which are secreted as exosomes from CML cells, stimulate HUVECs in order to increase the expression of intercellular adhesion molecule-1 (ICAM-1) and vascular cell adhesion molecule-1 (VCAM-1). This is one example of how CML cells can further their own progress by changing their preferred bone marrow microenvironment through exosome-mediated angiogenesis. In the treatment of ALL, the use of central nervous system (CNS)-transferring chemotherapy as a prophylaxis has been incorporated into treatment regimens. The CNS remains a critical site of invasion and recurrence that determines the prognosis in ALL. Ichiko et al. reported that in B-cell ALL, leukemic cell-derived exosomes containing IL-15 are taken up by astrocytes and brain vascular endothelial cells, leading to the production of vascular endothelial growth factor-AA (VEGF-AA) by astrocytes and to the disruption of the blood–brain barrier, resulting in central nervous system infiltration [73].

The evasion of immune responses by tumors has been attracting significant attention in recent years in terms of the development of new therapies such as immunotherapy, and the mechanism underlying immune response evasion in leukemia is becoming clearer. Exosomes have an important role in this mechanism as well [74,75]. Hong et al. reported that the serum of patients with acute myeloid leukemia contained more exosomes compared to healthy subjects, and it was particularly rich in Transforming Growth Factor (TGF)-β1, which plays an important role in the immune evasion of tumors [76]. TGF-β1 in leukemic cell-derived exosomes suppresses the tumor recognition function of NKG2D expressed on NK cells and cytotoxic T cells [77]. Methods to activate immune responses against tumor cells through exosomes, which have important functions in tumor immunity, could emerge as among the most important therapeutic targets in leukemia.

Exosomes have become increasingly important not only for their role in disease progression but also as biomarkers in early diagnosis at the time of initial onset or recurrence, as well as in prognosis prediction. Koolivand et al. reported that miR-155 levels were significantly higher in AML patients compared to controls. miR-155 may be regarded as a prognostic biomarker for AML, one that is noninvasive compared to the conventional bone marrow test [78].

Bernardi et al. found that *Breakpoint cluster region-Abelson (BCR-ABL)1* transcripts are detectable in CML cell-derived exosomes [79]. CML is a clonal myeloproliferative neoplasm caused by abnormal pluripotent hematopoietic stem cells carrying the BCR-ABL1 fusion caused by reciprocal translocation between chromosomes 9 and 22 and formation of the Philadelphia chromosome-positive (Ph^+^) CML [80]. In this case, conventional chemotherapy was not a curative treatment, and allogeneic HSCT remained the only possibility for a cure, although the number of cases who could undergo HSCT was limited. However, in the 2000s, tyrosine kinase inhibitors (TKIs) targeting the BCR-ABL1 p210 protein dramatically changed the prognosis of CML patients. TKI is an innovative molecular targeted therapy that inhibits the progression of CML. Indeed, it has achieved major or deep molecular responses (DMRs) in about 80% to 90% of patients [81,82]. In order to accurately monitor the effects of treatment, a test that can measure MRD, a small amount of which remains in the patient’s body, is required, and the BCR-ABL1 mRNA test has become increasingly popular worldwide. In particular, this test has been established as an International Scale (IS) to promote the standardization of measurement methods and values, and monitoring by IS is now recommended in the guidelines of various countries [83,84]. However, molecular relapse is experienced in more than 50% of patients undergoing TKI discontinuation, even after achieving deep and undetectable MRD as measured by the BCR-ABL1 mRNA test [85,86,87]. This suggests that the currently used MRD measurement may be insufficient as it is unable to detect leukemic cells lurking in the bone marrow microenvironment. By measuring the expression of BCR-ABL in CML cell-derived exosomes, it could be possible to evaluate residual CML cells in the bone marrow microenvironment, thus revealing the potential role of exosomes as biomarkers.

Another significant role of exosomes concerns the in vivo delivery of drugs, microRNA, and other molecules. Due to their noninvasive nature and easy accessibility, as well as the fact that microRNAs can evade destruction in protected spaces such as exosomes, the latter have emerged as a potentially new therapeutic strategy.

The role of exosome-derived leukemia has been extensively studied. Exosomes play various roles in the progression of leukemia, including regulation of the bone marrow microenvironment and angiogenesis; inhibition of hematopoiesis, metastasis to other organs, and immune escape; and transformation of the microenvironment into a tumorigenic microenvironment. Thus, an in-depth understanding of the broad roles played by leukemic exosomes will lead to a better understanding of the pathogenesis of leukemia and will yield important new perspectives not only concerning the development of new therapies to eliminate relapse but also early diagnosis and early detection of relapse.

### 3.2. Bone Marrow Remodeling by Leukemic Exosomes

Leukemic cell-derived exosomes play an important role in the remodeling of the bone marrow microenvironment [88] by regulating the communication between leukemic cells and the bone marrow microenvironment (Figure 2).

Exosomes also play an important role in altering the hematopoietic niches into leukemic cell niches in which normal hematopoiesis is difficult to establish [89]. Kumar et al. found that exosome-derived acute myeloid leukemia cells inhibit the normal function of bone marrow-derived mesenchymal stem cells (BMMSCs), thus facilitating remodeling of the bone marrow niche into a more desirable niche for leukemia progression [90]. Normal hematopoiesis is regulated by Dickkopf-1 (DKK-1). AML cell-derived exosomes suppress HSC support factors such as CXCL12 and Insulin-like Growth Factor-1 (IGF-1) in BM stromal cells by promoting DKK-1 expression, thereby reducing hematopoietic potential. Gao et al. found that in CML, tumor-suppressive microRNAs, especially the miR-320 secreted by leukemic cells, is taken up by adjacent BM-MSCs via heterogeneous nuclear ribonucleoprotein A1 (HNRNPA1) and significantly inhibit osteogenesis, remaking it into a bone marrow niche favorable for CML progression [91]. Umetsu et al. reported that AML cells and endothelial cells communicate via miRs secreted as exosomes [92]. When AML cell-derived mi-R92a was co-cultured with human umbilical vein endothelial cells, the expression of integrin α5, which is a target gene of miR-92a, was significantly downregulated in HUVECs, while miR-92 enhanced endothelial migration and tube formation, leading to remodeling of the BM.

Zhang et al. reported that miR-126 is closely related to the disease progression of CML [93]. In their study, miR-126 regulated CML progression by mediating the interaction between endothelial cells (ECs) and leukemic stem cells (LSCs) in the BM niches of CML patients. ECs secreted exosomes containing miR-126 to LSCs, which support the quiescence, self-renewal, and engraftment capacity of LSCs and that are involved in CML progression. The inhibition of BCR-ABL by TKIs caused an undesirable increase in miR-126, leading to enhanced LSC quiescence and persistence. This suggests that TKI-only treatment may lead to residual LSCs in the BM, which could lead to relapse following the discontinuation of TKI. Adding an inhibitor of miR-126 to TKI may offer a new therapeutic strategy for eradicating LSCs in CML patients. In addition to miR-126, miR-92 plays an important role in angiogenesis in the bone marrow of CML patients. It has been reported that such pro-angiogenic miRs increase the microvessel density and caliber in the bone marrow of CML patients. This remodeling of the bone marrow by leukemia-derived exosomes directly correlates with poor prognosis in CML patients [71,94].

Recent reports have revealed that long-term hematopoietic stem cells (LT-HSCs) are selectively quiesced and preserved in the bone marrow of leukemia patients [90,95,96]. Abdelhamed et al. have reported the mechanism by which leukemia-derived exosomes confer quiescence on residual HSCs in leukemic niches [97]. AML-EV transfers miR-1246 to LT-HSC, causing the translational suppression of the mammalian target of rapamycin (mTOR) subunit Raptor, suppressing protein synthesis and triggering their quiescence. Dormant HSCs are more susceptible to DNA damage, and DNA-damaged HSCs pose a significant risk for the development of additional mutations. Moreover, the acquisition of HSC mutations can lead to clonal expansion, malignant transformation, and relapse [98,99].

These studies suggest that leukemic cells interact with various other cells in the bone marrow microenvironment via exosomes to promote leukemic cell proliferation, acquire resistance to chemotherapy, and/or to support metastasis to distant organs by remodeling the microenvironment to their favorable advantage. These mechanisms are currently under investigation, but many aspects are still unknown. Conventional chemotherapy is associated with adverse events due to its invasiveness not only to leukemic cells, but also to normal cells. A therapeutic strategy focused on leukemic cells and their exosome-mediated bone marrow remodeling could represent a promising new strategy to improve treatment safety and even eradicate leukemia.

## 4. Conclusions

Here, we have summarized examples of leukemia–microenvironment interactions as well as the roles played by leukemia exosomes in remodeling bone marrow niches.

Thanks to the efforts of our predecessors, the relapse rate of leukemia has decreased compared to the past decade, due to the development of new therapeutic agents, the revision of treatment regimens, and the establishment of HSCT. Nonetheless, relapse remains a significant problem in leukemia because of the poor prognosis after relapse since conventional chemotherapy targets only leukemic cells, and in addition to its limited efficacy, serious adverse events that sometimes lead to treatment-related death are often a problem. Novel therapies that target not only leukemic cells, but also the bone marrow microenvironment, which is intimately involved in the biosynthesis of leukemic cells, may overcome drug resistance mechanisms. In this way, the myelosuppression that results from the disruption of normal hematopoiesis could be avoided, thereby leading to a therapeutic strategy with fewer adverse events. In addition, an exosome-based testing tool has the potential to identify subtle residual diseases that were previously undetectable. Indeed, using this tool to evaluate treatment and check for relapse could improve the prognosis of leukemia patients by targeting deeper treatment effects and identifying relapse earlier.

Elucidating the concept of leukemia–microenvironment interactions and the roles played by leukemia exosomes in remodeling bone marrow niches is expected to play a very important role in overcoming long-standing problems faced by leukemia patients worldwide.

## Figures and Tables

**Figure 1 ijms-22-01881-f001:**
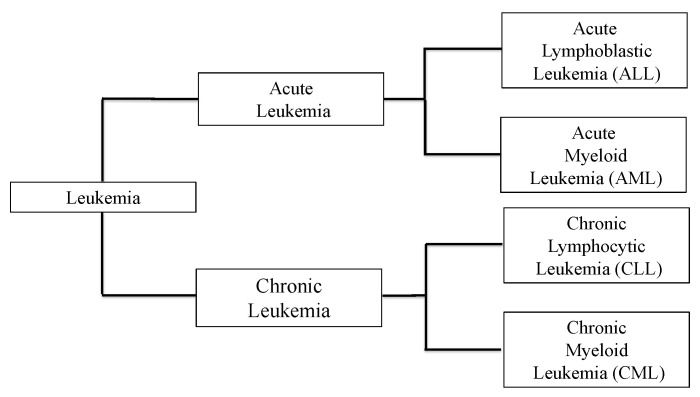
Four main types of Leukemia. Leukemia is a malignant clonal disease that originates from bone marrow hematopoietic stem cells. The four main types are acute lymphocytic leukemia (ALL), acute myeloid leukemia (AML), chronic lymphocytic leukemia (CLL), and chronic myeloid leukemia (CML). These are only the main examples of leukemia; there are multiple other types. A more detailed listing can be found in the revision of the 2016 World Health Organization (WHO) classification of myeloid neoplasms and leukemia.

**Figure 2 ijms-22-01881-f002:**
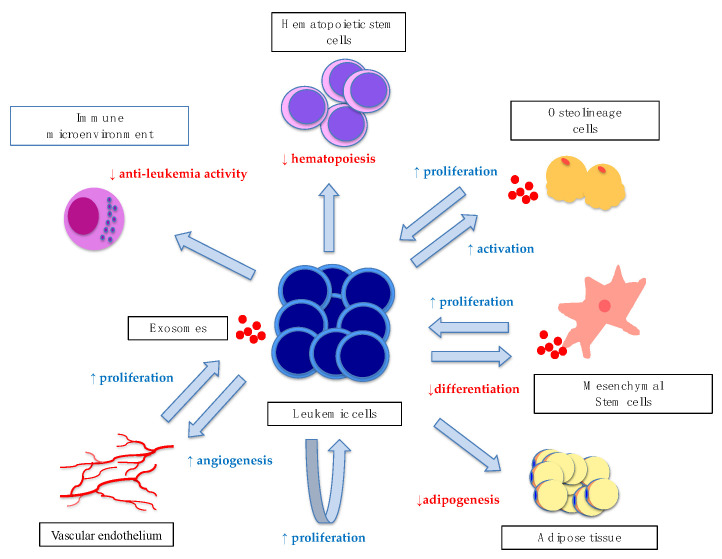
Interactions between leukemic exosomes and their niches. Shown are the multiple regulatory mechanisms implicated in promoting leukemic cell survival. Those that promote the function of the target cells (blue) and those that inhibit it (red) are depicted, respectively. Leukemia cells promote their own proliferation by the exosomes they secrete. The exosomes derived from leukemic cells inhibit the normal hematopoiesis of hematopoietic stem cells (HSCs), inhibit the differentiation of mesenchymal stromal cells (MSCs), and promote their own proliferation. In addition, they promote osteolineage cell activity as well as their own proliferation. Closer to the vascular endothelium, they promote angiogenesis, which allows for their own proliferation and migration to other organs. In the case of adipose tissue, exosomes interfere with adipogenesis and alter their functionality as a source of energy for themselves. Exosomes also inhibit the function of the normal hematopoietic pool.

## Data Availability

Data sharing not applicable.

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
