# Peer review of "Remodeling of Bone Marrow Niches and Roles of Exosomes in Leukemia"

_ijms, 2021, doi:10.3390/ijms22041881_

Round 1
Reviewer 1 Report
This paper claims to review the role of leukemic exosomal miRNAs and integrins in remodeling the bone marrow.
The paper starts with an abstract - an abstract has the goal to give a synopsis of the paper's content. Here the abstract is not focused and does not cover the expected content based on the title.
The introduction covers leukemia, exosomes and integrines and is well cited. However, the rest of the paper seems to be a collection of all what has been written on bone marrow niche and leukemia, and goes on and on about CXCR4, E Selectins and gap junction and Connexin, but nothing on integrins.
Then in line 407-410, there are some far reaching assumptions made regarding targeting integrins as a novel approach to treat leukemia (based on the paragraph above that describes exosomal integrins and its function in T cells of the GUT - but not a word about exosomal integrins in leukemia).
WHile the topic might be worthwhile a review paper - this paper does not cover the title. Either the authors have to change the title (and leave exosomes out of it all togehter) - or change the content of the paper to focus on three words : LEUKEMIA - EXOSOMES and EXOSOMAL INTEGRINS. In that case, the paper carries lost of superfluous data NOT related to the content as suggested by the title.
Reviewer 2 Report
Pag 1: row 43 ref 1 is not enough, add some on other types of leukemia
Pag 2: row 53: although later on, in the specific paragraph, various examples are cited please add references about exosomes associated with leukemia not with glioblastoma. Row 65: add ref. Row 75: ref 10 is not about leukemia, 11 is about glioblastoma, and 12 is about breast cancer
Pag 3: row 109: the cited references explain a correlation between the sympathetic nervous system and HSC ageing but these works are not about leukemic cells disrupting SNS functions
Pag 4: ref 41 is missing
Pag 6: row 230 This sentence is to assertive being only a hypothesis so should be rephrased: “…could be possible to evaluate…”. Row 235 did the authors really mean diagnostic strategy or they meant therapeutic strategy?
Pag 7: row 276-288 It is not clear to me how are these concepts related to exosomes, please explain better and the same for row 289 to 310
Pag 9: Row 419: check “as."
Author Response
We are grateful for the reviewers’ thoughtful comments and suggestions. As outlined below, we have addressed and/or clarified each of the issues previously raised in a point-by-point fashion. We believe that these changes have greatly improved the quality and significance of our study, and that the manuscript is now suitable for publication in Biomedicines (please note that corrections/changes throughout the revised manuscript appear highlighted in yellow).
R: Pag 1: row 43 ref 1 is not enough, add some on other types of leukemia
C: In accordance with the reviewer’s suggestion, we have added prognostic information for ALL and AML, as well as the corresponding references to the revised manuscript.
(see page 1, lines 55- 56)
(references 2,3,4)
R:Pag 2: row 53: although later on, in the specific paragraph, various examples are cited please add references about exosomes associated with leukemia not with glioblastoma.
C: As the reviewer suggested, we have added to the revised manuscript a reference relating to leukemia in.
(see page 3, line 89)
(reference 12)
R: Row 65: add ref.
C: As the reviewer suggested, we have added to the revised manuscript a reference related to leukemia.
(see page 3, line 101)
(reference 18)
R:Row 75: ref 10 is not about leukemia
C: As the reviewer suggested, we have added to the revised manuscript a reference related to leukemia.
(see page 3, line 112)
(reference 19)
R:ref 11 is about glioblastoma
C: As the reviewer suggested, we have added to the revised manuscript a reference related to leukemia.
(see page 3, line 110)
(reference 20)
R:Ref 12 is about breast cancer
C: As the reviewer suggested, we have added to the revised manuscript a reference related to leukemia.
(see page 3, line 110)
(reference 21)
R:Pag 3: row 109: the cited references explain a correlation between the sympathetic nervous system and HSC ageing but these works are not about leukemic cells disrupting SNS functions
C: We agree with the reviewer’s suggestion. We have changed the sentence ‘sympathetic neuropathy causes changes in the hematopoietic HSC niche and promotes leukemia progression’, as well as the corresponding reference.
(see page 4, lines 149-153)
(reference 32,33)
R:Pag 4: ref 41 is missing
C: We would like to thank this reviewer for his/her suggestion. We have cited the appropriate reference in the relevant sentence in the revised manuscript.
R: Pag 6: row 230 This sentence is to assertive being only a hypothesis so should be rephrased: “…could be possible to evaluate…”.
C: We agree with the reviewer’s suggestion. We have rephrased ‘could be possible to evaluate’.
(see page 7, line 333)
R: Row 235 did the authors really mean diagnostic strategy or they meant therapeutic strategy?
C: We agree with the reviewer’s suggestion. We meant ‘therapeutic strategy’ and we have changed from the wording from ‘diagnostic’ to ‘therapeutic’ in the revised manuscript.
(see page 8, line 339)
R:Pag 7: row 276-288 It is not clear to me how are these concepts related to exosomes, please explain better and the same for row 289 to 310
C: As the reviewer suggested, this is about the leukemic bone marrow niches.
We have replaced those sentences from the ‘Leukemia exosome’ chapter to the ‘Leukemic bone marrow niche’ chapter in the revised manuscript.
row 276-288 (see page 5, lines 232-243)
row 289 to 310. (see page 5, lines 209-217)
R:Pag 9: Row 419: check “as."
They should check this sentence because
there is a dot after “as" and then a capital letter. I suppose this is
just a typing mistake since I don’t think the sentence can end with "as":
C: We would like to thank this reviewer for pointing out this error as well as for the prompt response to the additional question. We have changed from ‘as. Conventional’ to ‘since conventional’ in the revised manuscript.
(see page 9, line 416)
Reviewer 3 Report
The authors offer a concise but comprehensive review on the role of exosomes in the biology of leukemia, specifically the bone marrow environment. I have some comments that need to be addressed before publication:
- The impact and understability of the review would be significantly improved by a figure, summarizing the most important functions of exosomes in the bone marrow niche.
- In the introduction, the authors write: Leukemia is a hematological malignancy originating from bone marrow hematopoietic stem cells, which can be broadly classified into chronic myelogenous leukemia (CML), chronic lymphocytic leukemia (CLL), acute myelogenous leukemia (AML), and acute lymphocytic leukemia (ALL) based on tumor origin and clinical course (Fig. 1). This is a very "undergraduate" way of looking at leukemia. There are many more types of leukemia, the 4 types discussed by the authors are just the 4 main types of leukemia.
- Some parts of the review are unconnected, stand-alone sentences for which the authors can improve their relation with the rest of the manuscript text, for example:
- The percentage of fatty cells in the bone marrow increases with age, and this fatty tissue-rich bone marrow microenvironment is considered to be suitable for leukemia cell survival [27].
- Other cells such as osteoblasts, platelets, and macrophages also play important roles in maintaining HSCs, but leukemia cells disrupt these interactions, suppressing normal hematopoiesis and creating a favorable environment for their own growth [28-31].
- The bone marrow microenvironment protects leukemia cells against chemotherapeutic drugs [35] and constitutes a potential therapeutic target of leukemia
- Leukemia cells disrupt the normal bone marrow niche and transform it into a more convenient niche for themselves so that they can grow more favorably than normal hematopoietic stem cells [18]. This is an important biological feature of leukemia cells and has also be called "hijacking of bone marrow niches", which was described in more detail in 10.1016/j.bbcan.2017.03.010
- The bone marrow niche is divided into the endosteal niche and the vascular
niche, which are known to work in concert with the HSC [15,16]. Some authors have also described the reticular bone marrow niche. In addition, it has been proposed that there is only one type of bone marrow niche, which consists of a periarteriolar compartment and a perisinusoidal compartment 10.1016/j.bbcan.2017.03.010. This is a minor nuance but it may be an important one, given the focus of the review.
Author Response
We are grateful for the reviewers’ thoughtful comments and suggestions. As outlined below, we have addressed and/or clarified each of the issues previously raised in a point-by-point fashion. We believe that these changes have greatly improved the quality and significance of our study, and that the manuscript is now suitable for publication in Internal Journal of Molecular Sciences (please note that corrections/changes throughout the revised manuscript appear highlighted in yellow).
C: The impact and understability of the review would be significantly improved by a figure, summarizing the most important functions of exosomes in the bone marrow niche.
R: As the reviewer suggested, we have added a figure summarizing the important functions of exosomes in the bone marrow niche.
(see page 8, Figure 2)
C: In the introduction, the authors write: Leukemia is a hematological malignancy originating from bone marrow hematopoietic stem cells, which can be broadly classified into chronic myelogenous leukemia (CML), chronic lymphocytic leukemia (CLL), acute myelogenous leukemia (AML), and acute lymphocytic leukemia (ALL) based on tumor origin and clinical course (Fig. 1). This is a very "undergraduate" way of looking at leukemia. There are many more types of leukemia, the 4 types discussed by the authors are just the 4 main types of leukemia.
R: We agree with the reviewer’s suggestion. We have changed the wording from ‘broadly classified’ to ‘four main types of leukemia’ in the revised manuscript. (see page 2, lines 47-49)
In addition, we have commented on the 2016 revised WHO classification in revised manuscript. (see page 2, lines 43-47). We have also added a comment regarding the classification on Figure 1. (see page 2, lines 65-66)
C:Some parts of the review are unconnected, stand-alone sentences for which the authors can improve their relation with the rest of the manuscript text, for example:
The percentage of fatty cells in the bone marrow increases with age, and this fatty tissue-rich bone marrow microenvironment is considered to be suitable for leukemia cell survival [27].
R: We agree with the reviewer’s suggestion.
We have added a description on adipocytes and leukemia bone marrow niches in the revised manuscript. (see page 4, lines 154-162)
C:Other cells such as osteoblasts, platelets, and macrophages also play important roles in maintaining HSCs, but leukemia cells disrupt these interactions, suppressing normal hematopoiesis and creating a favorable environment for their own growth [28-31].
R: We agree with the reviewer’s suggestion. We have added a description regarding those cells and leukemia bone marrow niches in the revised manuscript. (see page 4, lines 138-153)
C:The bone marrow microenvironment protects leukemia cells against chemotherapeutic drugs [35] and constitutes a potential therapeutic target of leukemia
R: We agree with the reviewer’s suggestion. We have separated this section into paragraphs with an improved and more coherent flow from one subject to the next. (see page 5, lines 219- 221)
C:Leukemia cells disrupt the normal bone marrow niche and transform it into a more convenient niche for themselves so that they can grow more favorably than normal hematopoietic stem cells [18]. This is an important biological feature of leukemia cells and has also be called "hijacking of bone marrow niches", which was described in more detail in 10.1016/j.bbcan.2017.03.010
R: We would like to thank this reviewer for his/her constructive feedback.
We have added the sentence ‘Leukemia cells hijack bone marrow niche’ as well as the corresponding references. (see page 3, lines 116- 123) (reference 11)

R:The bone marrow niche is divided into the endosteal niche and the vascular
niche, which are known to work in concert with the HSC [15,16]. Some authors have also described the reticular bone marrow niche. In addition, it has been proposed that there is only one type of bone marrow niche, which consists of a periarteriolar compartment and a perisinusoidal compartment 10.1016/j.bbcan.2017.03.010. This is a minor nuance but it may be an important one, given the focus of the review.
R: We would like to thank this reviewer for his/her constructive feedback.
We had added the description of the reticular bone marrow niche, as well as comments on the two compartments in the revised manuscript. (see page 2, lines 74 - page 3 line 87)
Round 2
Reviewer 3 Report
The improvements now support publication.